# The Toll-Like Receptor 5 agonist flagellin prevents *Non-typeable Haemophilus influenzae*-induced infection in cigarette smoke-exposed mice

Magdiel Pérez-Cruz[1]ᵒ, Bachirou Koné[1]ᵒ, Rémi Porte[1], Christophe Carnoy[1], Julien Tabareau[1], Pierre Gosset[2], François Trottein[1], Jean-Claude Sirard[1], Muriel Pichavant[1]ᵒ, Philippe Gosset[1]ᵒ*

1 Univ. Lille, CNRS UMR9017, Inserm U1019, CHRU Lille, Institut Pasteur de Lille, CIIL—Center for Infection and Immunity of Lille, Lille, France, 2 Service d'Anatomo-pathologie, Hôpital Saint Vincent de Paul, Lille, France

ᵒ These authors contributed equally to this work.

* philippe.gosset@pasteur-lille.fr

**Data Availability Statement:** All relevant data are within the manuscript and its Supporting Information files.

## Abstract

Chronic obstructive pulmonary disease (COPD) is a major cause of morbidity and mortality worldwide. The major bacterial cause of COPD exacerbations is non-typeable *Haemophilus influenzae* (NTHi). 25 to over 80% of cases are associated with NTHi. This susceptibility to infection involves a defective production of interleukin (IL)-22 which plays an important role in mucosal defense. Prophylactic administration of flagellin, a Toll-like receptor 5 (TLR5) agonist, protects healthy mice against respiratory pathogenic bacteria. We hypothesized that TLR5-mediated stimulation of lung immunity might prevent COPD exacerbations. Mice chronically exposed to cigarette smoke (CS), which presented COPD symptoms, were infected with NTHi and intraperitoneally treated with recombinant flagellin following a prophylactic or therapeutic protocol. Compared with control, cigarette smoke-exposed mice treated with flagellin showed a lower bacterial load in the airways, the lungs and the blood. This protection was associated with an early neutrophilia, a lower production of pro-inflammatory cytokines and an increased IL-22 production. Flagellin treatment decreased the recruitment of inflammatory cells and the lung damages related to exacerbation. Morover, the protective effect of flagellin against NTHi was altered by treatment with anti-IL-22 blocking antibodies in cigarette smoke-exposed mice and in *Il22*⁻/⁻ mice. The effect of flagellin treatment did not implicated the anti-bacterial peptides calgranulins and defensin-β2. This study shows that stimulation of innate immunity by a TLR5 ligand is a potent antibacterial treatment in CS-exposed mice, suggesting innovative therapeutic strategies against acute exacerbation in COPD.

## Introduction

Chronic obstructive pulmonary disease (COPD) is characterized by a progressive and irreversible decline in lung function [1]. Being the third leading cause of death worldwide, it is mainly

**Funding:** This work was granted by the Conseil Régional du Nord-Pas de Calais [StreptoCOPD project; grant number # 13005300]. This work was supported by the Institut National de la Santé et de la Recherche Médicale (Inserm), the Centre National de la Recherche Scientifique (CNRS) and the University of Lille. Funders had no role in study design, data collection, data analysis, interpretation, and writing of the report.

**Competing interests:** The authors have declared that no competing interests exist.

caused by chronic exposure to cigarette smoke (CS) or pollutants [2]. Inhalation of CS essentially leads to activation of epithelial cells and macrophages responsible for the mobilization of effector and immuno-modulatory cells including neutrophils and natural killer T (NKT) cells [3,4]. The chronic inflammatory response progressively leads to airway remodeling, impaired bacterial clearance and parenchymal destruction in the lungs, further culminating in irreversible airflow limitation [5] as experienced in our murine model of chronic exposure to CS. These components are involved in the increased susceptibility of COPD patients to bacterial and viral airway infections.

Airway colonization with bacteria such as *Haemophilus influenzae*, *Streptococcus pneumoniae* and *Moraxella catarrhalis* contributes to the pathogenesis and clinical course of the disease [6]. This colonization is responsible for lung infection leading to exacerbations of the disease, which have a strong impact on health status, exercise capacity, lung function, and mortality. Non-typeable *Haemophilus influenzae* (NTHi), a Gram-negative coccobacillus that lacks a polysaccharide capsule, is an important cause of COPD exacerbations and comorbidity [7,8]. Acute exacerbations in patients invariably scarred the chronic course of COPD [9]. Bacterial infections are first controlled by the innate immune system, which implicated pathogen-associated molecular pattern (PAMP) recognition by Toll-like receptors (TLR) such as those recognizing flagellin (TLR5) responsible for the mobilization of effector cells [10]. During COPD, bacterial infection is characterized by an increased influx of immune cells, including neutrophils, macrophages, dendritic cells (DC) and T lymphocytes [3,11,12]. However, this response is not effective enough to clear the pathogens. In this context, we recently reported a defective production of IL-22 in response to bacteria both in COPD patients and mice chronically exposed to CS, whereas IL-17 production is only altered after infection by *S. pneumoniae* [13,14]. Interestingly, the Th17 cytokines IL-17 and IL-22 promote the recruitment of neutrophils, the synthesis of antimicrobial peptides and the expression of tight junction molecules [15,16], a mechanism explaining the essential role of IL-22 in the clearance of NTHi [14]. Morover, supplementation of COPD mice with recombinant IL-22 increases the clearance of the bacteria and prevents the development of COPD exacerbations in mice. However, IL-22 expression is also promoted by exposure to CS and is involved in COPD pathogenesis [17]. Several reports showed that activation of innate receptors, including TLR, is able to elicit protective immune responses against infections [18,19]. Among them, systemic administration of flagellin, the main component of bacterial flagella and the TLR5 ligand, induces immediate production of Th17 cytokines through the activation of DC and type 3 innate lymphoid cells [20].

In this study, we hypothesized that systemic administration of recombinant form of flagellin could inhibit the development of NTHi-induced COPD exacerbation episodes through an appropriate protective IL-22 response. We reported here that systemic stimulation of the innate immunity by flagellin from *Salmonella enterica* serovar Typhimurium (FliC) prevents COPD exacerbation induced by NTHi. We also showed that the protective effect of flagellin against NTHi is dependent of IL-22 but was not associated with the modulation of calgranulins (S100A8/S100A9) and defensin-β2.

## Material and methods

### Animals

Male C57BL/6 (WT) or IL-22$^{-/-}$ C57BL/6j mice of both sexes, 6–8 weeks old were obtained from Janvier Labs (Le Genest-St-Isle, France) or Jean-Christophe Renauld (Brussel, Belgium), respectively. WT mice were daily exposed to cigarette smoke (CS) during 12 weeks (5 cigarettes/day, 5 days/week during 12 weeks) to induce COPD pathogenesis [4]. They were

exposed in a whole-body chamber integrated to the Inexpose system (EMKA, Paris-France). Research cigarettes 3R4F were obtained from the University of Kentucky Tobacco and Health Research Institute (Lexington, KY, USA). The control group was exposed to ambient air. After 12 weeks of CS or air exposure, mice were either treated intranasally with phosphate buffered saline (PBS) or NTHi (n = 4 per group), three days after the last exposure to CS. *Il22*⁻/⁻ mice were not exposed to CS before before infection with NTHi and controls received PBS. All procedures were performed according to the Pasteur Institute, Lille, Animal Care and Use Committee guidelines (agreement number N°AF16/20090). The present project has been approved by the national Institutional Animal Care and Use Committee (CEEA 75) and received the authorization number APAFIS# 7281.

## Mice infection and flagellin treatment

NTHi 3224A strain was grown to log-phase in brain-heart infusion (BHI) broth (AES Laboratory) supplemented with 10μg/ml haematin and 10μg/ml nicotinamide adenine dinucleotide (NAD) (SIGMA, St Louis, MI, USA), and stored à -80°C in BHI 10% glycerol for up to 3 months.

For mouse infection, working stocks were thawed, washed with sterile PBS, and diluted to the appropriate concentration. The number of infectant bacteria was confirmed by plating serial dilutions onto chocolate agar plates. Mice were anesthetized and intranasally (i.n.) infected with $2.5 \times 10^6$ CFU of NTHi.

For preparation of heat-killed (HK) NTHi, bacteria were grown to a log-phase ($O.D_{600nm}$ = 0.7–0.8 units) and inactivated at 56°C for 1 hour in a hot-water-bath. Broth cultures were then plated onto chocolate agar plates and incubated overnight to check bacterial inactivation.

Native flagellin was purified and depleted in endotoxin as described previously [21]. To evaluate the prophylactic effect, 5μg of flagellin was administrated intraperitoneally (i.p.) just before bacterial challenge. In some experiments, we evaluated a therapeutic protocol in which flagellin was intraperitoneally injected 6h after the infection. For IL-22 neutralizing experiment, mice received 200μg of neutralizing anti-IL-22 (AM22) or control isotype (a mouse IgG2a) antibodies intravenously 5 minutes before infection.

## Sample collection and processing

Mice were sacrificed 24h and 48h post-infection by NTHi. Broncho-alveolar Lavage (BAL) fluids, lungs, spleen and blood were collected and kept on ice till the processing or immediately frozen in liquid nitrogen.

BAL was performed by instilling 5 x 0.5 ml of sterile PBS + 2% fetal bovine serum (FBS) via a 1 ml sterile syringe with 23-gauge lavage needle into a tracheal incision. BAL samples were used for cytokine analysis, flow cytometry analysis and numbering of CFUs. Lung tissues were collected aseptically and analyzed for CFU counts, cytokines analysis, histology and pulmonary cell analysis (flow cytometry analysis and lung cell restimulation). For this, lungs were perfused with PBS and the left lobe was treated with collagenase (Sigma-Aldrich). The leucocyte-enriched fraction was collected using a Percoll gradient (GE Healthcare) before flow cytometry staining and culture. Blood was used for the determination of CFU counts and measurement of cytokine concentrations.

## Flow cytometry

Cells harvested from BAL and lungs were washed and incubated with antibodies (BD, Franklin lakes, NJ, USA) for 30 min in PBS before being washed. Staining was performed as described in online supplementary information. Data were acquired on a LSR Fortessa (BD Biosciences)

and analyzed with FlowJo™ software v7.6.5 (Stanford, CA, USA). Gating strategies are previously reported by Sharan et al. [22]. Debris were excluded according to size (FSC) and granularity (SSC). Immune cells expressing CD45 were gated to analyse frequency, activation and number of cell subsets. Phenotypes are shown in the Table 1.

## Cytokine measurement

Levels of IFN-γ, IL-1β, IL-6, IL-17, IL-22, IL-23 and tumor necrosis factor alpha (TNF-α) were quantified in blood, lung tissue lysates and BAL using commercial ELISA kits (Invitrogen, San Diego, USA; Biotechne, Minneapolis, USA) (Table 2). In addition, defensin-β2 concentrations were also measured in lung extracts and BAL by ELISA (Abbexa, Cambridge, UK). Similarly, levels of IFN-γ, IL-17, and IL-22 were measured in the supernatants of dissociated lung cells ($0.5 \times 10^6$ of cells) alone or re-stimulated with HK NTHi during 72h.

## RT-PCR quantification of mRNA expression

Quantitative RT-PCR was performed to quantify mRNA of interest (Table 3). Results were expressed as mean ± SEM of the relative gene expression calculated for each experiment in folds ($2^{-\Delta\Delta Ct}$) using *Gapdh* as a reference, and compared to non-infected PBS-treated control mice.

## Histological analysis

To study lung remodeling post-infection with NTHi, lungs were inflated and fixed in formalin. The lungs were then paraffin-embedded; cross-sections were cut and stained with hematoxylin and eosin. To define the lung lesions we have used a histopathologic score quantifying lung injury and including both lung remodeling and inflammation (Table 4). More specifically, this scoring includes the Extent of lung injury, the alveolar wall thickness, the presence of hyaline membrane, the neutrophilic alveolitis, the bronchial epithelial degeneration, the neutrophilic and lymphocytic peribronchitis, the vasculitis, the emphysema and the hemorrage for a cumulative score from 0 to 30. The evaluation was blindly performed. In order to evaluate emphysema, we measured mean linear intercept on photos from lung sections by using Image J software (NIH). Results were expressed as pixels (mean ± SD)

## Statistical analysis

The data are expressed as mean ± SEM. Results were statistically analyzed using one way anova analysis (Kruskal Wallis test) followed by Dunn's multiple comparison test (PRISM software, v5 GraphPad)), and expressed in terms of probability (p). Differences were considered significant when $p < 0.05$ (*: $p < 0.05$; **: $p < 0.01$; ***: $p < 0.001$).

**Table 1. Phenotype of the major cell populations identified in this report.**

| Cell population | Phenotype |
|---|---|
| Alveolar Macrophages | F4/80$^+$ CD11c$^+$ CD64$^+$ SiglecF$^+$ |
| Neutrophils | F4/80$^-$ CD11c$^-$ CD11b$^+$ Ly6G$^+$ |
| Dendritic cells | F4/80$^-$ CD11c$^+$ I-Ab$^+$ CD64$^-$ |
| Inflammatory monocytes | F4/80$^+$ CD11c$^-$ Ly6G$^-$ Ly6C$^+$ CCR2$^+$ |
| Conventional T cells | CD5$^+$ TCRαβ$^+$ NK1.1$^-$ |
| NKT like cells | NK1.1$^+$ TCRαβ$^+$ |

**Table 2. List of the antibodies and of the ELISA kits used in this study.**

| Flow cytometry mAb | Target | Manufacturer | Catalog Nb |
|---|---|---|---|
| | FITC- I-Ab | Miltenyi Biotech | 130-102-168 |
| | PE-F4/80 | Miltenyi Biotech | 130-102-422 |
| | PerCP-Cy5.5—CD103 | BD Biosciences | 563637 |
| | PE-Cy7—CD11c | BD Biosciences | 558079 |
| | APC—CCR2 | Miltenyi Biotech | 130-119-658 |
| | AF700—CD86 | BD Biosciences | 560581 |
| | APC-H7- Ly6G | BD Biosciences | 560600 |
| | V450—CD11b | BD Biosciences | 560455 |
| | VioGreen—CD45 | Miltenyi Biotech | 130-110-665 |
| | BV605—Ly6C | Biolegend | 128036 |
| | BV786—CD64 | BD Biosciences | 741024 |
| | PE-CF594—SiglecF | BD Biosciences | 562757 |
| | FITC—CD5 | Miltenyi Biotech | 130-102-574 |
| | Tetramer mCD1d 167ms | NIH facility | 30663 |
| | PerCP-Cy5.5—NK1.1 | Miltenyi Biotech | 130-103-963 |
| | PE-Cy7—CD4 | Miltenyi Biotech | 130-102-411 |
| | APC—CD25 | Miltenyi Biotech | 130-102-550 |
| | AF700—CD69 | BD Biosciences | 561238 |
| | APC-Vio770—TCRγδ | Miltenyi Biotech | 130-104-016 |
| | VioBlue -TCRβ | Miltenyi Biotech | 130-104-815 |
| | V500—CD8 | BD Biosciences | 130-109-252 |
| | BV605—CD45 | Biolegend | 103140 |
| **ELISA kits** | **Target** | **Manufacturer** | **Catalog Nb** |
| | IFN-γ ELISA kit | Invitrogen | 88-7314-88 |
| | IL-1β Duoset | Biotechne | DY401 |
| | IL-6 ELISA kit | Invitrogen | 88-7064-88 |
| | IL-17 ELISA kit | Invitrogen | 88-7371-88 |
| | IL-22 Duoset | Biotechne | DY582 |
| | IL-23 ELISA kit | Invitrogen | 88-7230-88 |
| | TNF-α ELISA kit | Invitrogen | 88-7371-88 |
| | Defensin-β2 | Abbexa | Abx254734 |

## Results

### Intraperitoneal administration of flagellin accelerates the clearance of NTHi in CS-exposed mice

Mice chronically exposed to CS developed the major COPD features [4] and were intranasally challenged with NTHi and previously treated or not with flagellin (FliC) before infection (Fig 1A). As we previously reported [22], the bacterial load was higher in CS-exposed mice infected with NTHi (in the BAL and lung tissue at 24h and in the BAL at 48h) (Fig 1B) than in infected control mice. Intraperitoneal injection of FliC significantly enhanced 24h post-infection (p.i.), the clearance of NTHi in BAL, lungs (Fig 1B) and the blood (S1 Fig) from CS-exposed mice compared to the PBS-treated mice. At 48h p.i., treatment with flagellin decreased the bacterial load in the BAL but not in the lungs and blood at this time point.

Since COPD exacerbation is associated with an altered immune cell response relative to control mice [13,23], we next characterized immune cells in the lungs and BAL. At 24h p.i., we observed an increased total cell number in the BAL and lungs of infected CS-exposed mice,

**Table 3. Primer sequences for qRT-PCR in mice.** Forward (F) and reverse (R) primers are cited.

| Genes | | Sequences |
|---|---|---|
| Gapdh | F | TGCCCAGAACATCATCCCTG |
| | R | TCAGATCCACGACGGACACA |
| Defβ2 | F | AAAGTATTGGATACGAAGCAGAACTTG |
| | R | GGAGGACAAATGGCTCTGACA |
| Defβ3 | F | TGAGGAAAGGAGGCAGATGCT |
| | R | GGAACTCCACAACTGCCAATC |
| Camp | F | CAGAGCGGCAGCTACCTGAG |
| | R | TCACCACCCCCTGTTCCTT |
| S100a8 | F | TGTCCTCAGTTTGTGCAGAATATAAA |
| | R | TCACCATCGCAAGGAACTCC |
| S100a9 | F | CACCCTGAGCAAGAAGGAAT |
| | R | TGTCATTTATGAGGGCTTCATTT |
| Reg3b | F | ATGCTGCTCTCCTGCCTGATG |
| | R | CTAATGCGTGCGGAGGGTATATTC |
| Reg3g | F | CTGTGGTACCCTGTCAAGAGC |
| | R | GGCCTTGAATTTGCAGACAT |

compared to controls (Fig 2A). An increased number of neutrophils, alveolar macrophages (AM) and dendritic cells (DC) (p<0.05) in the BAL were reported of infected CS-exposed mice compared to uninfected mice, whereas neutrophils and DC were higher in the lungs (Figs 1D and S2). This increase was consistent at 48h p.i. for the total cell number and the neutrophil count (S2C Fig, p<0.01). After treatment with FliC, the total cell number was reduced in both control and CS-exposed mice infected with NTHi. This was related in CS-exposed mice to a lower number of neutrophils in the BAL, and a trend for AM and DC. DC activation evaluated in the airways by the expression of CD86 and the MHC molecule I-Ab was not modulated in CS-exposed mice treated with FliC (S2B and S2D Fig). Regarding lymphocytes, we showed a significantly higher recruitment of both NKT and T cells in the BAL (p<0.05) and the lungs upon infection of controls and CS-exposed mice (Figs 1D and S2A). Treatment with

**Table 4. Lung injury scoring criteria.**

| Lung injury | Score | | | | |
|---|---|---|---|---|---|
| Scale | 0 | 1 | 2 | 3 | 4 |
| Extent of lung injury | Absence | <25% | 26 to 50% | 51 to 75% | >75% |
| Alveolar wall thickness | ≤ 1 rbc | > 1 ≤ 2 rbc | 3 to 5 rbc | 6 to 10 rbc | > 10 rbc |
| Hyaline membranes | Absence | Presence | NA | NA | NA |
| Neutrophilic alveolitis | Absence | <10 neutrophils/HPF | 10 to 20/HPF | 21 to 50/HPF | > 50/HPF |
| Suppuration | Absence | Presence | NA | NA | NA |
| Bronchial epithelial degeneration | Absence | Presence | NA | NA | NA |
| Neutrophilic peribronchitis | Absence | <10 neutrophils/HPF | 10 to 20/HPF | 21 to 50/HPF | > 50/HPF |
| Lympho-hiostiocytic peribronchitis | Absence | <10 mononuclear cells/HPF | 10 to 20/HPF | 21 to 50/HPF | > 50/HPF |
| Vasculitis (inflammation) | Absence | Presence | NA | NA | NA |
| Vasculitis (necrosis) | Absence | Presence | NA | NA | NA |
| Emphysema | Absence | < 25% | 25 to 50% | to 75% | > 75% |
| Hemorrhage | Absence | Presence | NA | NA | NA |

rbc: Red blood cells; NA: Not applicable; HPF: High Power Field (magnification x250).

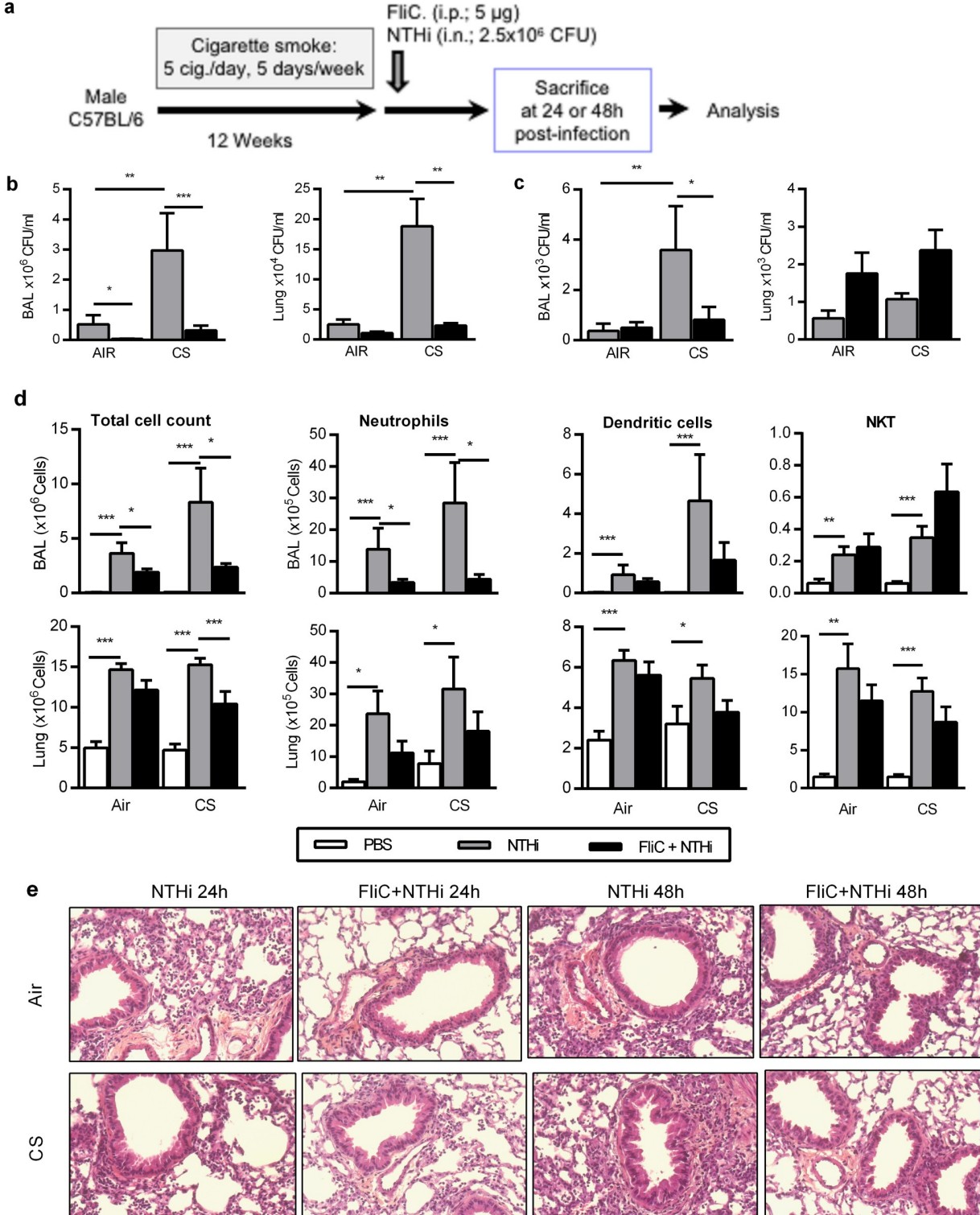

**Fig 1. Treatment with Flagellin prevents the NTHi-induced COPD exacerbation in CS-exposed mice.** (a) To assess the impact of flagellin treatment on COPD exacerbation by NTHi, mice were chronically exposed to cigarette smoke during 12 weeks followed by intranasal challenge with NTHi 2.5x10[6] CFU and flagellin administration (5 μg; i.p.). Mice were euthanized at 24h or 48h after NTHi challenge for analysis of (b) Colony Forming Unit (CFU) counts in Broncho-alveolar lavage fluid (BAL) and lungs. (c) The total number of recruited cells as well as the absolute number of neutrophils, dendritic cells and NKT was reported in BAL and lungs of control *versus* COPD mice infected or not with NTHi and treated or not with flagellin. The samples were collected 24h after NTHi challenge. (d) Lung histopathology was performed on

control and CS-exposed mice injected with PBS or FliC and infected or not with NTHi at 24h and 48h after challenge. Three independent experiments have been performed with 4 mice in each group. Data are expressed as mean ± SEM. *: p<0.05, **: p<0.01, ***: p<0.001.

FliC strongly reduced the number of T lymphocytes at both day 1 and 2 p.i. whereas the activation of these cells was not modulated in comparison with infected mice as evaluated by CD69 expression was not changed (S2B Fig).

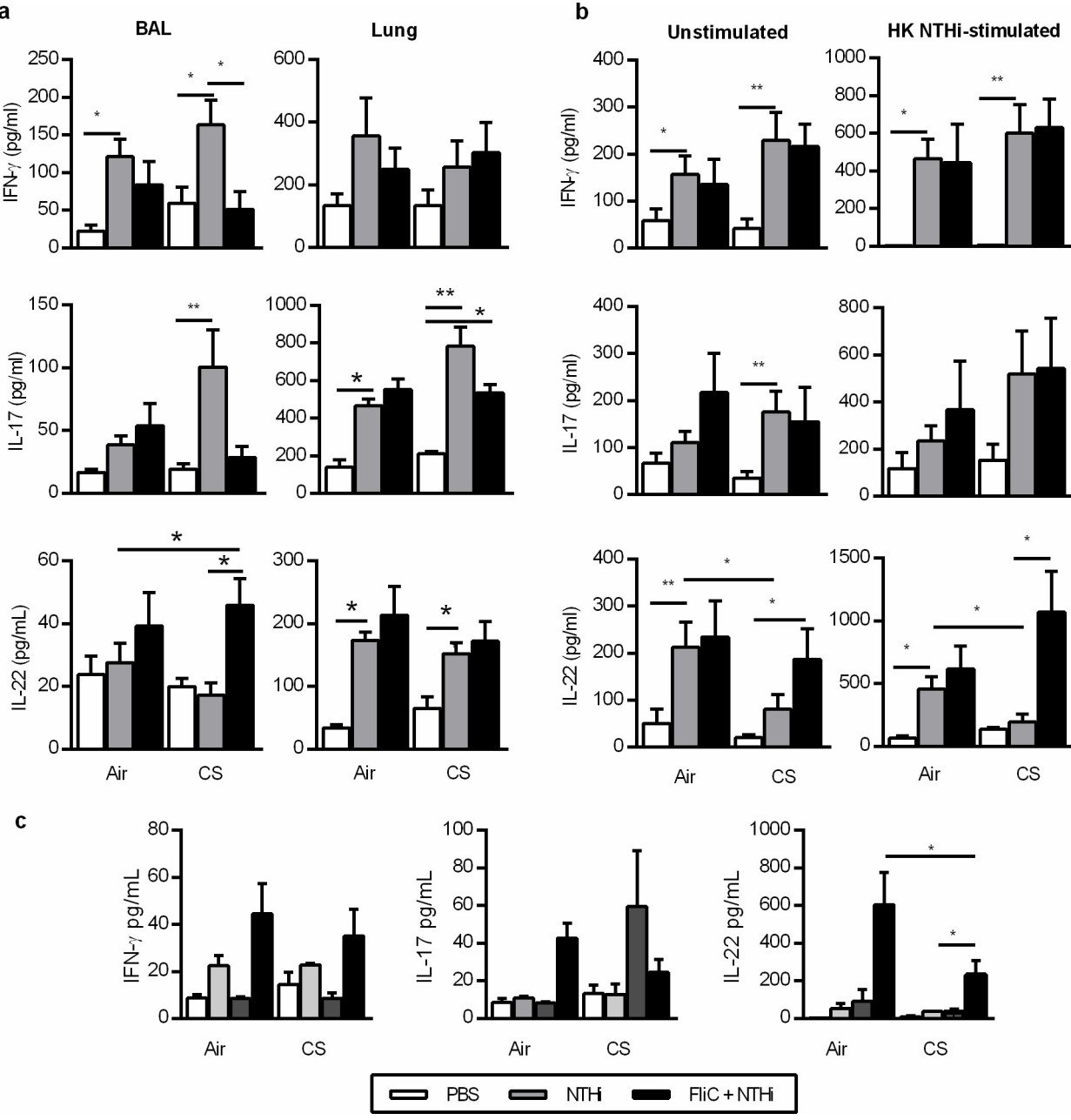

**Fig 2. Flagellin limits NTHi induced inflammation in CS-exposed mice.** (a) The concentrations of IFN-γ, IL-17 and IL-22 were analyzed in the BAL and the lungs of control and CS-exposed mice injected with PBS or FliC and infected or not with NTHi at 24h after NTHi challenge. (b) These cytokines were also evaluated in supernatants of lung cells either unstimulated or in vitro restimulated with heat-killes NTHi. (c) The concentrations of IFN-γ, IL-17 and IL-22 were measured in the sera at 24h after NTHi challenge. Three independent experiments have been performed with at least 3 mice in each group. The data are expressed as mean ± SEM. *: p<0.05, **: p<0.01, ***: p<0.001.

Histopathological analysis of lung tissues showed that infection with NTHi in CS-exposed mice induced more inflammation and remodeling tissue than in control mice (Figs 1D and S2). Histopathologic analysis confirmed that treatment with FliC markedly decreased inflammatory cell recruitment both in peribronchial and alveolar spaces of NTHi-infected CS-exposed mice at 48h p.i. compared to control mice. Moreover, infected CS-exposed mice exhibited some features of pneumonia with alveolitis and a strong vasculitis 48h p.i. whereas these lesions were not observed in animals treated with FliC. This was confirmed by the histopathological score evaluating both inflammation and lung tissue remodeling ($7 \pm 0.41$ vs $4.33 \pm 0.31$ in PBS- and FliC-treated infected CS-exposed mice, respectively, $p < 0.05$). Whereas the MLI was increased in not infected CS-exposed mice as compared with Air mice (S3 Fig), infection by NTHI decreased the MLI only at day 1p.i. in CS-exposed mice. Treatment with FliC also reduces the emphysema at day 2 p.i.

Altogether, these data demonstrated that treatment with FliC amplified the clearance of NTHi in CS-exposed mice, a result associated with a lower lung inflammation and less damage mostly due to the infection.

## Flagellin treatment modifies the cytokine response consecutive to NTHi infection within the lung of CS-exposed mice

Since Th1 and Th17 cytokines are involved in the control of lung inflammation and bacterial infection, we analyzed their concentrations as well as those of cytokines involved in their production in the BAL and the lung lysates. Significantly higher levels of IL-17 (Fig 2A), IL-1β, IL-6 and TNF-α (S4A Fig) were detected in BAL and lung tissue lysates from infected CS-exposed mice, compared to non-infected animals at 24h after infection. IL-22 production was only increased in lung lysates and there was no difference between Air and CS-exposed mice (Fig 2A). Interestingly, treatment with FliC significantly reduced the concentrations of IFN-γ, IL-1β, and TNF-α in the BAL (S4A Fig; $p \leq 0.05$). There was no effect in control mice. Treatment with flagellin significantly increased the production of IL-22 in the BAL but not in the lung of CS-exposed mice at day 1 p.i. (Fig 1C) as well as IL-23 levels (p = NS, S4B Fig). To further analyze the potential of lung immune cells to promote efficient antibacterial immune response, these cells were restimulated *ex vivo* with heat-killed (HK) bacteria and their ability to produce cytokine profiles were assessed. Infection with NTHi increased the production of IFN-γ, IL-17, and IL-22 in Air-mice as compared with PBS mice at 24h p.i. (Fig 2B). In infected CS-exposed mice, higher levels of IFN-γ, IL-17 were detected compared to the controls whereas the levels of IL-22 were not induced by the infection in both unstimulated and HK NTHi stimulated lung cells ($p < 0.05$). In CS-exposed mice treated with FliC, lung cells produced more IL-22 than PBS-treated infected mice ($p < 0.05$) whereas the levels of IL-17 and IFN-γ remained unchanged. There was no difference in control mice.

In parallel, we also analyzed the production of these cytokines in the blood. Infection by NTHi did not modulate the blood concentration of IFN-γ and IL-22 whereas it tended to increase the levels of IL-17 in CS-exposed mice (Fig 2C). Interestingly, FliC significantly increased the concentrations of IL-22 in CS-exposed mice.

The FliC-induced protection was associated with a lower pro-inflammatory cytokine burst in the lung and an increased IL-22 production both in the lung and the blood of CS-exposed mice.

## Flagellin increases the ability of spleen T cells to produce IL-22

Since FliC was administered intraperitonally, we evaluated its impact on the immune cell phenotype within the spleen. We did not detect a significant modification of the number of

the major APC in flagellin-treated and infected mice compared to only infected mice as illustrated for inflammatory monocytes and cDC2 (S5 Fig). In addition, we observed no statistical modulation of the expression of I-Ab in both cell types. Regarding T lymphocytes, their absolute number was not significantly modulated in both Air and CS-exposed mice after infection but also after treatment with FliC as shown for iNKT cells and T CD8+ cells. Similarly, treatment with FliC did not significantly amplify CD25 expression on both cell types in both Air- and CS-exposed mice. Production of IFN-γ, IL-17 and IL-22 was measured in supernatants of total spleen cells. Infection by NTHi and treatment with Flagellin had no effect on the production of these cytokines in unstimulated cells (Fig 3A). Administration of FliC significantly increased the ability of spleen cells from Air mice to produce IL-17 whereas it significantly amplify the IL-22 production in mice exposed to CS after stimulation by NTHi (Fig 3B).

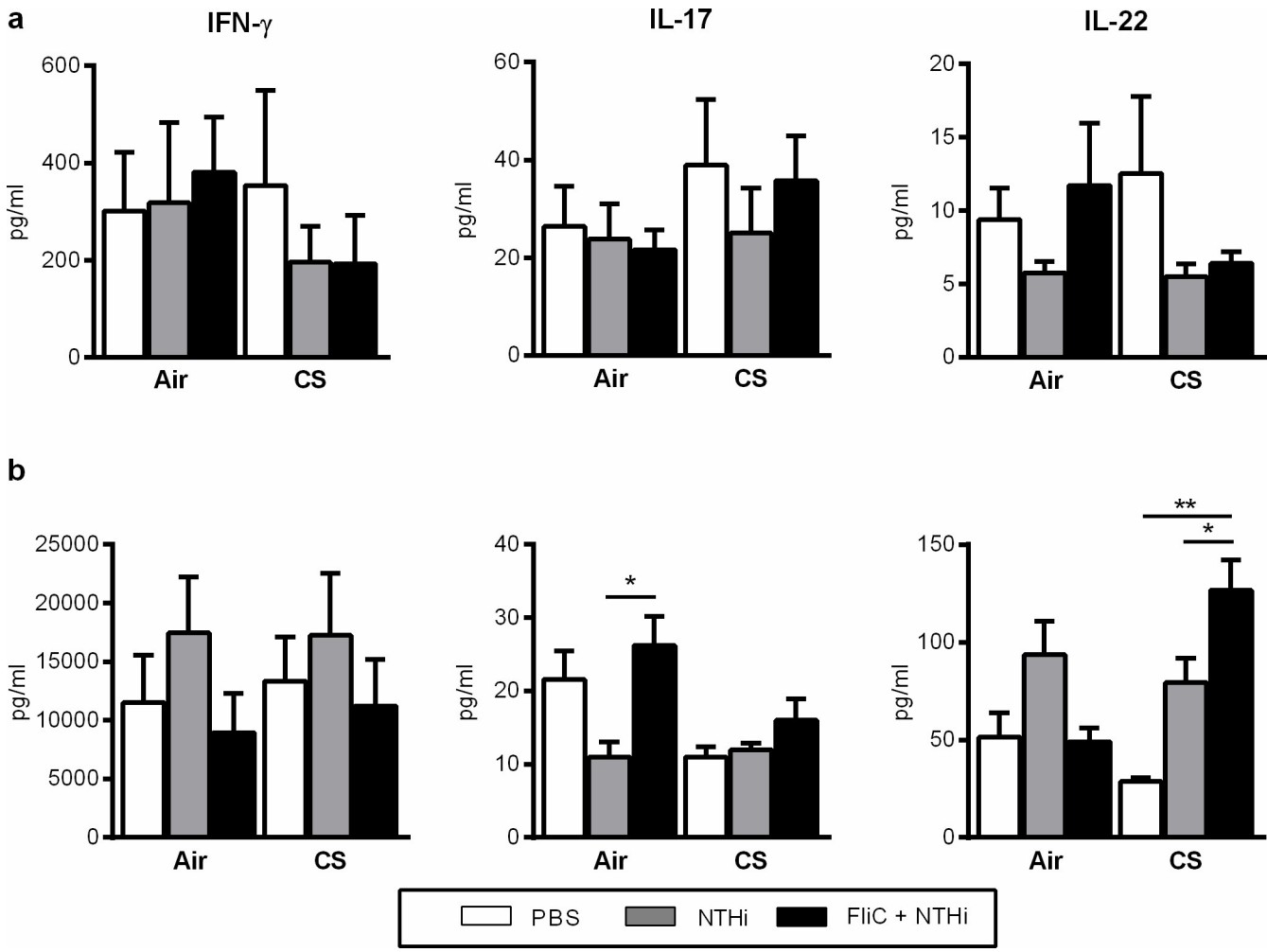

**Fig 3. Flagellin modulate the production of IL-22 cytokines in splenocytes from NTHi-infected cigarette smoke-exposed mice.** (a) IFN-γ, IL-17 and IL-22 concentrations were measured in the supernatants of unstimulated spleen cells of mice infected or not with NTHi and treated or not with FliC, at 48h after infection. (b) IFN-γ, IL-17 and IL-22 concentrations were measured in the supernatants of activated spleen cells of mice infected or not with NTHi and treated or not with FliC, at 48h after infection. Spleen cells were activated by addition of heat-killed-NTHi (MOI 10) during 48 hours. Two independent experiments have been performed with at least 3–4 mice in each group. *: $p < 0.05$ and **: $p < 0.01$.

## IL-22 is important for Flagellin-mediated protection in COPD exacerbation by NTHi

It has already been described that the prophylactic effect of flagellin against lung infection by *S. pneumoniae* is mediated by early (between 2 and 24h) overexpression of IL-22, in a TLR5 dependent manner, through the increase of IL-22+ ILC3 in the lung [20]. In order to confirm the implication of IL-22 in the effect of flagellin, we first treated *Il22*-/- mice with this TLR5 ligand before infection by NTHi (Fig 4A). Compared to wild type (WT) mice, *Il22*-/- mice cleared NTHi within comparable timing although the bacterial load was higher in mice (Fig 4B). Treatment with FliC did not significantly decrease the bacterial load in the BAL and lung compared to PBS treated *Il22*-/- mice whereas it did in WT mice (Fig 1B). Regarding the inflammatory cell influx, the administration of FliC did not modulate the absolute number of neutrophils, AM and DC in the BAL or in the lung of *Il22*-/- mice (Fig 4B and 4C). The levels of IL-17 and IFN-γ were also measured and we did not detect a significant effect of FliC on the concentration of these cytokines in *Il22*-/- mice except in the BALF at 48h p.i. (S6 Fig) as shown in WT mice. Histological analysis showed that the lesions in infected *Il22*-/- mice have the same intensity as in infected WT mice (Figs 1D and 4D). Moreover, FliC slightly limited the lung remodeling in infected *Il22*-/- mice but with a lower degree than in WT mice. In contrast with the data obtained in WT mice (Fig 1D), although the inflammatory infiltrate persists in both PBS and FliC-treated NTHi-infected *Il22*-/- mice (histologic score: 7.1 ± 0.48 versus 4.77 ± 0.75, respectively).

To further investigate the implication of IL-22 in the protective effect of flagellin, WT CS-exposed mice were intravenously treated with anti-IL-22 blocking antibodies before FliC treatment and NTHi infection (Fig 4E). 24h after infection, anti-IL-22 completely abrogated the effect of FliC on the bacterial load reduction (Fig 4F). Treatment with the blocking antibody and FliC did not affect the number of neutrophils and DC counts in BAL and lung compared with infected CS-exposed mice (Fig 4G) whereas it slightly increased the absolute number of macrophages within the lung.

These results demonstrate that the protective effect of flagellin during COPD exacerbation is at least partly dependent of IL-22.

## Impact of flagellin on the production of anti-microbial peptides in infected CS-exposed mice

Flagellin as well as IL-22 cytokine are also able to promote the production of antibacterial peptides [24,25]. To investigate this pathway, we analyzed the expression of anti-microbial peptide mRNA in the lung and the protein level of S100A8 and S100A9 in the BAL. The delay in NTHi clearance observed in CS-exposed mice compared to control was not associated with significant changes in the expression of *Defb2*, *S100A8*, *S100A9* (Fig 5A), *Defb3*, *Reg3g* (S7 Fig), as compared to infected control mice. Administration of flagellin increased the levels of *Defb2* in CS-exposed mice whereas it decreased the mRNA expression of *Defb3* and *Reg3g* in CS-exposed mice. In order to confirm these data at the protein level, we measured the protein concentration of defensin-β2, S100A8 and S100A9 (Fig 5B). Infection with NTHi markedly increased the concentrations of defensin-β2, S100A8 and S100A9 in the BAL of both control and CS-exposed mice. Treatment with FliC did not modulate the levels of S100A8 and S100A9 in both groups of mice at both 24 and 48h after infection. Upregulation of *Defb2* mRNA was not associated with an increased concentrations of defensin-β2 in both BALF and lung extracts in both Air and CS-exposed infected mice.

These data showed that the effects of FliC are not associated with the upregulation of defensin-β2 and calgranulins synthesis.

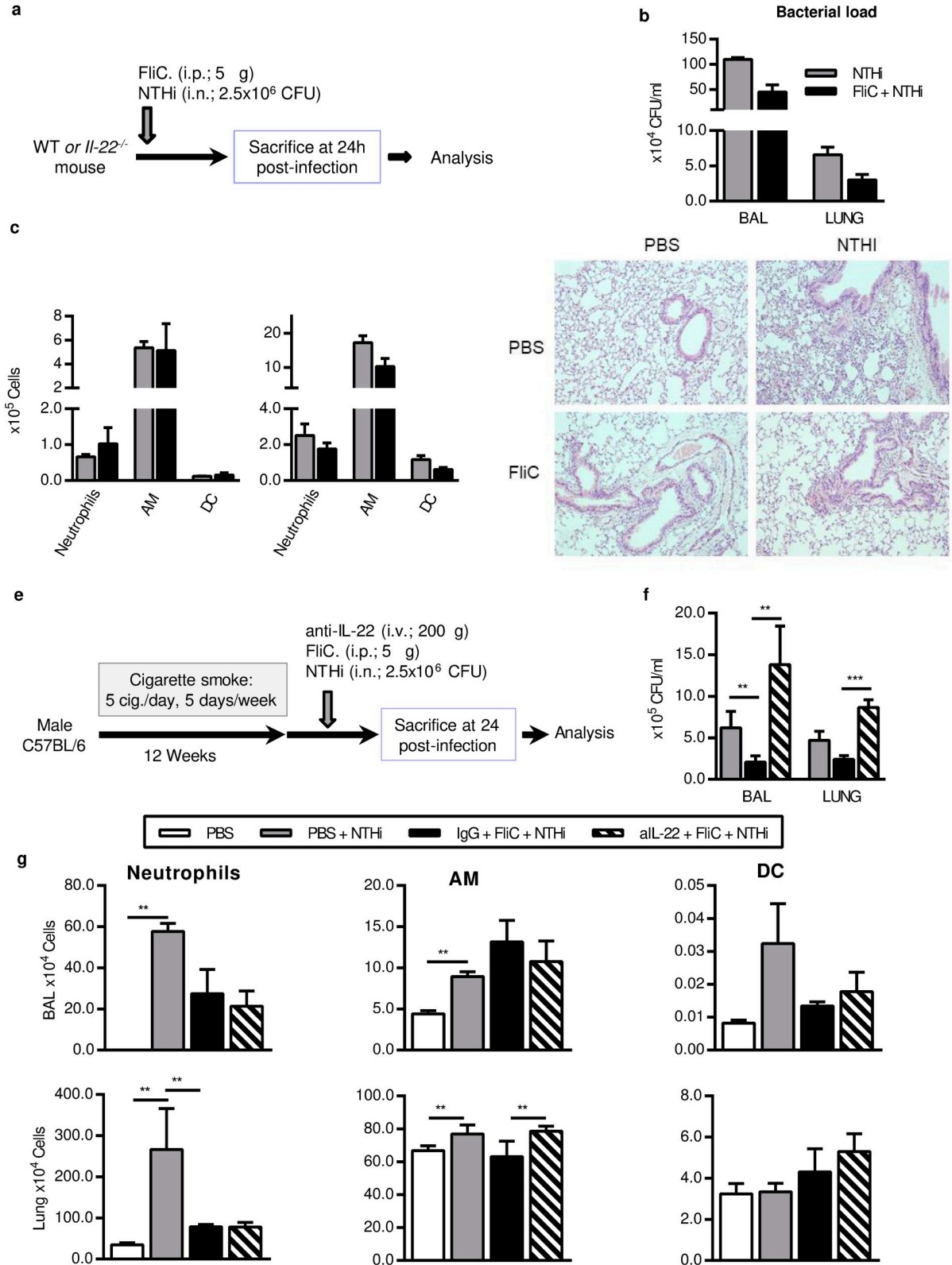

**Fig 4. IL-22 is important in flagellin-mediated protection during NTHi infection.** (a-d) Infection in *Il22*[-/-] mice, (e-g) treatment of COPD mice with anti-IL-22 antibody. (a) To identify the role of IL-22 in the effect of flagellin during NTHi infection, *Il22*[-/-] mice were challenged with NTHi at 2.5x10**6** CFU for 24h. (b) Bacterial load was assessed in BAL and lungs. (c) Absolute number of neutrophils, alveolar

macrophages (AM) and dendritic cells (DC) in BAL and lungs of mice infected with NTHi. (d) Lung histopathology was performed in *Il22*[-/-] mice injected with PBS or FliC and infected or not with NTHi. The data are expressed as mean ± SEM of 3 independent experiments (*n*≥3). (e) Wild type CS-exposed mice were intravenously injected with anti-IL-22 neutralizing antibodies 5min before FliC treatment and infected with NTHi for 24h. (f) CFU count, (g) Neutrophil numbers, AM and DC count were reported in the BAL and the lungs. The data are expressed as mean ± SEM of 2 independent experiments (*n*≥3). **: p<0.01, ***: p<0.001.

## Discussion

The increased susceptibility to infection during COPD is linked to a defect in IL-22 production related with an altered innate immune response [13,22,23]. In this study, we demonstrated that treatment with flagellin, a TLR5 ligand, is able to improve the ability of CS-exposed mice to clear bacteria including NTHi. The mechanism involved in this bacterial clearance is at least partially dependent of IL-22 but is not associated with an increased production of anti-microbial peptide defensin-β2 and calgranulins. Interestingly, the clearance of the bacteria was associated with a reduced inflammatory infiltrate and a decreased production of inflammatory cytokines in the lung of CS-exposed mice resulting in a less intense remodeling of lung tissues. Moreover, this treatment is also able to promote the NTHi-induced IL-22 production by PBMNC from healthy subjects [26].

The efficiency of FliC was demonstrated in an acute model of COPD exacerbation, using NTHi, reproducing most of the biological characteristics of this episode [3,27]. However, the increased inflammatory reaction associated with neutrophil and macrophage influx and the pro-inflammatory cytokine storm in CS-exposed mice did not allow to clear the bacteria. An altered production of IL-22 seems to be an essential mechanism responsible for this defect [13,22].

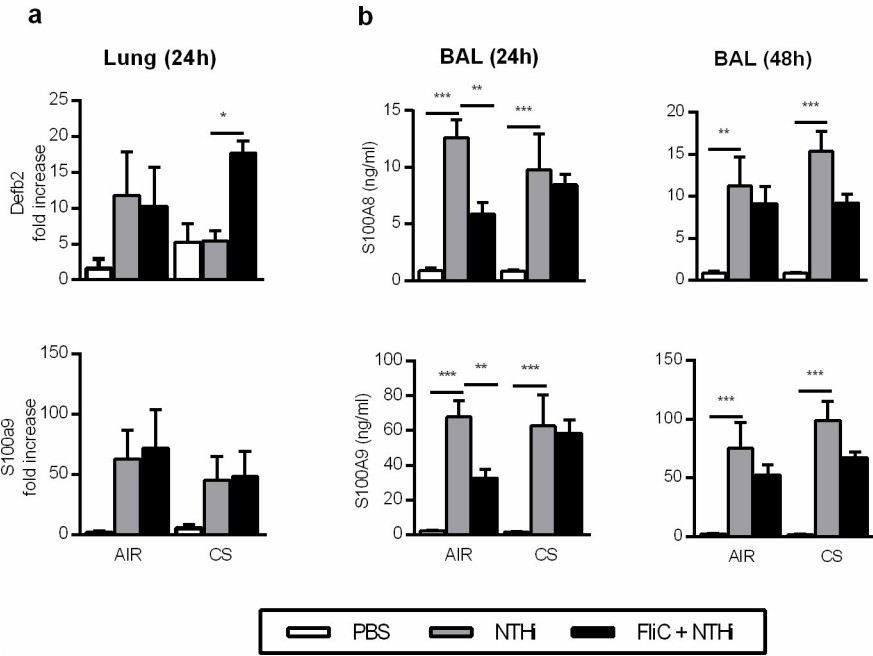

**Fig 5. Effect of flagellin on the production of anti-microbial peptides in CS-exposed mice infected with NTHi.** (a) mRNA expression of *Defb2*, *S100A8* and *S100A9* in the lungs of control *versus* CS-exposed mice infected or not with NTHi and treated or not with flagellin. (b) Concentrations of Defb2, *S100A8* and *S100A9* in the BAL of control *versus* CS-exposed mice infected or not and treated or not with flagellin. BAL were collected 24h and 48h after infection. Three independent experiments have been performed with at least 3 mice in each group. The data are expressed as mean ± SEM*: p<0.05, **: p<0.01, ***: p<0.001.

We are the first to demonstrate that treatment with FliC is also efficient against NTHi whereas its interest has been shown in infection with other bacteria including *S. pneumoniae* in non-CS-exposed mice [28–30]. Since the pathophysiology of COPD exacerbation episodes implicated a defect in IL-22 production and a deleterious effect of neutrophils on lung function, we choose to treat our mice by intraperitoneal route rather than a local administration which promotes a strong neutrophil recruitment in the airways. As previously reported with Sp in control (non COPD) mice [20,31,32] and in CS-exposed mice [26], the systemic treatment with FliC in NTHi-infected mice increases the IL-22 production in the BAL and in the supernatant of restimulated pulmonary cells without increase of the number and the activation of DC and AM in the airways. Nevertheless, we cannot exclude that the increased ability to produce IL-22 was linked to the recruitment of some populations of lymphocytes including T cells and NKT cells in CS-exposd mice. Both AM and DC expressed TLR5 and it has been reported that they are stimulated after administration of FliC [31,33]. We can suspect that FliC promotes the response to both Sp and NTHi not only in the lung but also in spleen and draining lymph nodes. Indeed, we detect a significant increase of IL-22 concentrations both in the blood and the supernatants of spleen cells from infected CS-exposed mice suggesting the circulation of immune cells and/or mediators between the spleen and the lung. The implication of IL-22 in the FliC-induced protection against NTHi was confirmed by the lack of bacterial decrease and the lower modulation of the inflammatory cell recruitment in *Il22*$^{-/-}$ mice. The role of IL-22 in the control of the bacterial load was confirmed by the pre-administration of neutralizing anti-IL-22 antibody in FliC-treated CS-exposed mice. These data are in line with our previous report showing that the supplementation with recombinant IL-22 is able to accelerate the clearance of the bacteria and to limit the consequences of bacterial infection in CS-exposed mice [13]. Interestingly, this is not associated with a modulation of IL-17 and IFN-γ in CS-exposed mice confirming that these cytokines are not essential for the clearance of NTHi [14,23].

Interestingly, we also described that the protection induced by prophylactic treatment with FliC was associated with a decrease in lung inflammatory cell recruitment and in airway remodeling in infected CS-exposed mice. This effect is probably the consequence of the accelerated clearance of the bacteria and/or to the activation of effector cells. IL-22 synthesis upregulation did not seem to be essential for the control of the inflammation since the inflammatory cell recruitment was not affected in FliC-treated *Il22*$^{-/-}$ mice compared to WT mice. Moreover, the production of IL-17 and/or IFN-γ is not essential in the anti-bacterial activity since flagellin treatment does not increase their production and IL-22 neutralization did not modulate their concentrations [15,16].

In order to determine how FliC increase the bacterial clearance, we analyzed the expression of AMP. FliC is known to promote the anti-microbial response as well as the production of chemokines and pro-inflammatory cytokines (including TNF-α and IL-1β) in airway epithelial cells, macrophages and neutrophils [32]. Through this mechanism, FliC might prime the anti-bacterial activity of effector cells including macrophages and neutrophils. We can also suspect that this treatment restores the barrier function of the airway mucosa since bacteria translocation within the blood is also decreased in CS-exposed mice (S1 Fig). Whereas our data show that the treatment with FliC promotes the production of calgranulins in Sp-infected mice [26], this is not the case after NTHi infection. It has been reported that S100A8 and 9 are induced by both SP- and NTHi-infection, and they are major players in the host response against pneumococcal infection by increasing lung recruitment of neutrophils and macrophages [34,35]. Their implication in the clearance of NTHi is unknown whereas the lack of effect after FliC treatment is probably related to the high level of induction by NTHi alone. In contrast, defensin-β2 is active against the major pathogens involved in COPD exacerbations including Sp and NTHi, while defensin-β1 appeared to only affect *M. catarrhalis* [36]. Recent findings show that

NOD2-mediated defensin-β2 production participates in the protection against NTHi-induced otitis [37]. Moreover, virus-induced altered expression of defensin-β results in an increased load of NTHi within the upper airways, which likely promotes development of lung infection [38]. However, the measurement of protein concentrations for these AMP did not reveal an increased production after FliC administration although we cannot exclude that this treatment induces the production of other AMP.

Interestingly, we also described that the protection induced by prophylactic administration with FliC was associated with a decrease in lung inflammatory cell recruitment and in airway remodeling in infected CS-exposed mice. Although these data must be confirmed in clinical practice, they suggest that this treatment can limit the consequences of bacterial infection during COPD, particularly the alteration of lung functions and the development of comorbidities. By restoring an efficient barrier, we can also hypothesize that this treatment will reduce the systemic inflammatory effects of the exacerbation. According to previous reports [20,29] and to our data with SP, we can also hypothesize that this treatment was also efficient through curative administration alone or in combination with antibiotics against NTHi. Since the safety of this adjuvant has been shown for clinical application (https://clinicaltrials.gov/ct2/show/results/NCT00966238), the interest of this treatment for COPD exacerbation might be predicted.

In conclusion, we demonstrated that treatment by flagellin is able to control bacterial infection in CS-exposed mice and to limit their consequences in terms of lung inflammation and remodeling. Although this effect seems to be partially dependent of the production of IL-22, we also suggest that the protection induced by FliC leads to the modulation of anti-microbial peptide production. FliC-induced restoration of an efficient bacterial clearance and limitation of the inflammatory reaction could be a step forward the treatment of COPD exacerbation.

## Supporting information

**S1 Fig. Treatment with Flagellin prevents the blood dissemination of NTHi in CS-exposed mice.**
(PDF)

**S2 Fig. Flagellin modulated the inflammatory cell recruitment whereas it did not affect their activation in NTHi-infected cigarette smoke-exposed mice.**
(PDF)

**S3 Fig. Flagellin reduce the emphysema in the lung of NTHi-infected cigarette smoke-exposed mice.**
(PDF)

**S4 Fig. Modulation by flagellin of cytokine production in the lung of Air- and cigarette smoke-exposed mice infected with NTHi.**
(PDF)

**S5 Fig. Flagellin did not affect the main immune cell populations nor their activation in the spleen of NTHi-infected cigarette smoke-exposed mice.**
(PDF)

**S6 Fig. Cytokine production in the lung of WT and IL-22-/- mice following flagellin treatment.**
(PDF)

**S7 Fig. Flagellin reduce the Defb3 and REG3g mRNA expression in the lung of NTHi-infected cigarette smoke-exposed mice.**
(PDF)

**S1 File.**
(ZIP)

## Acknowledgments

We gratefully acknowledge Eva Vilain and Gwenola Kervoaze for their excellent support in completion of experiments. We also acknowledge Dr Jean Christophe Renauld (Brussel, Belgium) which has generated the IL-22-deficient mice. We also thank Hélène Bauderlique for her help for advice on flow cytometry (BICel Cytometry Plateform, Institut Pasteur de Lille, France). A special thanks to François Trottein for critical reviewing of the paper.

## Author Contributions

**Conceptualization:** Magdiel Pérez-Cruz, Bachirou Koné, Christophe Carnoy, Jean-Claude Sirard, Muriel Pichavant, Philippe Gosset.

**Data curation:** Magdiel Pérez-Cruz, Bachirou Koné, Rémi Porte, Pierre Gosset, François Trottein, Muriel Pichavant, Philippe Gosset.

**Formal analysis:** Magdiel Pérez-Cruz, Bachirou Koné, Rémi Porte, Christophe Carnoy, Julien Tabareau, Pierre Gosset, François Trottein, Jean-Claude Sirard, Muriel Pichavant, Philippe Gosset.

**Funding acquisition:** Muriel Pichavant, Philippe Gosset.

**Investigation:** Magdiel Pérez-Cruz, Bachirou Koné, Rémi Porte, Christophe Carnoy, Julien Tabareau, Pierre Gosset, François Trottein, Philippe Gosset.

**Methodology:** Magdiel Pérez-Cruz, Bachirou Koné, Rémi Porte, Christophe Carnoy, Julien Tabareau, Jean-Claude Sirard, Philippe Gosset.

**Project administration:** Philippe Gosset.

**Resources:** Magdiel Pérez-Cruz, Bachirou Koné, Rémi Porte, Christophe Carnoy, Julien Tabareau, François Trottein, Jean-Claude Sirard.

**Supervision:** Philippe Gosset.

**Validation:** Magdiel Pérez-Cruz, Bachirou Koné, Rémi Porte, Pierre Gosset, Muriel Pichavant, Philippe Gosset.

**Visualization:** Magdiel Pérez-Cruz, Bachirou Koné, Pierre Gosset.

**Writing – original draft:** Magdiel Pérez-Cruz, Bachirou Koné, Rémi Porte, Christophe Carnoy, Julien Tabareau, Pierre Gosset, François Trottein, Jean-Claude Sirard, Muriel Pichavant, Philippe Gosset.

**Writing – review & editing:** Magdiel Pérez-Cruz, Bachirou Koné, Rémi Porte, Christophe Carnoy, Julien Tabareau, Pierre Gosset, François Trottein, Jean-Claude Sirard, Muriel Pichavant, Philippe Gosset.

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
