## [Decision Letter · Decision Letter 0]

13 Aug 2020

PONE-D-20-19459

The Toll-Like Receptor 5 agonist flagellin prevents Non-typeable Haemophilus influenzae-induced exacerbations in cigarette smoke-exposed mice

PLOS ONE

Dear Dr. Gosset,

Thank you for submitting your manuscript to PLOS ONE. After careful consideration, we feel that it has merit but does not fully meet PLOS ONE’s publication criteria as it currently stands. Therefore, we invite you to submit a revised version of the manuscript that addresses the points raised during the review process.

Both reviewers raised significant issues that must be considered and addressed carefully to allow further consideration of publication. Please note that methodological rigour is a prerequisite for publication in PloS One and therefore it is essential that comments related to these (group sizes, appropriate control groups, replication of findings) are addressed adequately.

We look forward to receiving your revised manuscript.

Kind regards,

Aran Singanayagam

Academic Editor

PLOS ONE

Additional Editor Comments: 

Both reviewers raised significant issues that must be considered and addressed carefully to allow further consideration of publication. Please note that methodological rigour is a prerequisite for publication in PloS One and therefore it is essential that comments related to these (group sizes, appropriate control groups, replication of findings) are addressed adequately.

Journal Requirements:

2. During our internal review, we noticed you have overlapping text with the dissertation of one of the authors published here:

https://tel.archives-ouvertes.fr/tel-01916088/document

Please clarify what type of copyright, if any, the dissertation has and who the copyright owner is. PLOS ONE cannot (re)publish material without sufficient permission from the original copyright holder to publish under a CC BY license. If the copyright is not owned by the author and is not under a under a CC BY license, please complete one of the following:

Please provide proof that the owner of the content (a) has given you written permission to use it, and (b) has approved of the CC BY license being applied to their content. You may have the following form completed by the owner as proof: https://journals.plos.org/plosone/s/file?id=7c09/content-permission-form.pdf.

Alternatively, you may electronically request permissions through from the copyright holder and send us proof of approval, as long as the approval clearly shows that the owner has approved of the CC BY license being applied to their content. Please see https://journals.plos.org/plosone/s/licenses-and-copyright for more information.

4. Your ethics statement must appear in the Methods section of your manuscript. If your ethics statement is written in any section besides the Methods, please move it to the Methods section and delete it from any other section. Please also ensure that your ethics statement is included in your manuscript, as the ethics section of your online submission will not be published alongside your manuscript.

Reviewers' comments:

Reviewer's Responses to Questions

**Comments to the Author**

1. Is the manuscript technically sound, and do the data support the conclusions?

Reviewer #1: No

Reviewer #2: Partly

2. Has the statistical analysis been performed appropriately and rigorously? 

Reviewer #1: No

Reviewer #2: No

3. Have the authors made all data underlying the findings in their manuscript fully available?

Reviewer #1: No

Reviewer #2: No

4. Is the manuscript presented in an intelligible fashion and written in standard English?

Reviewer #1: No

Reviewer #2: No

5. Review Comments to the Author

Reviewer #1: The manuscript describes use of a mouse model of chronic (12 week) CS-exposure induced lung disease directly followed by NTHi challenge.

The title is confusing – ‘…..haemophilus influenzae induced exacerbations in CS-exposed mice’. Exacerbations of what…? It suggests that there is a pre-existing lung disease and NTHi infection is making this worse ie exacerbation. However as I read on there is no evidence of established (12 week CS-induced) lung disease. So I think the title needs re-thinking to better reflect the model.

FLiC is given systemically (via intraperitoneal injection) – what is the rationale for this (to treat COPD exacerbation)? Have the authors tried targeted (airway) treatment?

Results

The first statement – ‘mice chronically exposed to CS developed the major COPD features’. What is the evidence for this? Simply providing a reference to another paper as evidence that CS exposure induced COPD features (which are not defined…) is not adequate. Based on title, abstract, intro this study appears (although a bit vague eg title) about bacterial exacerbation of COPD. Figs 1b and 1c present BAL and lung bacterial load data. Note the labelling is incorrect in figure legend. The results state levels of NTHi in the blood were measured – there is no data for this. The right panel for fig 1c indicates increased levels of bacteria in lung tissue with FliC treatment?

Please show individual data points for each graph -are you showing n = 4 mice for one experiment or combining three repeats to show n = 12 per treatment? If n = 4 data is analysed this is not parametric and therefore mean +/-SEM not appropriate.

Fig 1D does include a PBS treated group to enable comparison of CS vs air treated mice. There is no evidence of airway (BAL) inflammation. Airway inflammation woul be considered a ‘major feature of COPD’. Why do the authors think CS exposure has not caused airway inflammation? Fig 1E – no PBS group here so cannot determine if CS exposure (alone) vs air (alone) has caused any COPD-like disease such as alveolar destruction as measured by mean linear intercept via quantitation of lung histology.

The data in figure 1 is not consistent with a model of bacterial COPD exacerbation since there are essentially no disease outcomes apparent in the CS (PBS) group vs Air (PBS) group. Ie no evidence that CS exposure has caused disease. The lung disease outcomes presented are driven by NTHi. The possibility that prior CS exposure has modified susceptibility to NTHi induced lung disease. However, no statistical comparison of CS (NTHi) vs Air (NTHi) has been conducted so there is no evidence that CS exposure has modified response to NTHi.

Fig 1E – there is no CS (PBS) and Air (PBS) group so cannot assess CS-induced lung disease.

Fig 2 Unclear how many data points per treatment have been analysed. Please show individual data points – particularly if some groups have 3 mice so cannot determine if stats are appropriate… It is not clear to this reviewer what the key results here. Yes the different treatments are modifying cytokine expression – but the relevance of this to a specific pathway, disease mechanisms is not clear. Several sentences begin with ‘interestingly’ - interesting perhaps.. but is the relevance to bacterial COPD exacerbation, particularly given that this is not really modelling COPD exacerbation (no evidence of COPD-like disease).

Fig 3. A lot of sup data or data not shown. Not clear why you are looking at immune cells in spleen when this a study of lung infection/disease. 3A cytokine production by spleen cells from CS-exposed mice +/- NTHi +/- FliC. Again don’t know number of data points for this data, and not statistically significant so does not add anything. 3B – why is this relevant and how does it inform on pathogenesis bacterial COPD exacerbation?

Fig 4 ‘to identify the role of IL-22 in the effect of flagellin during NTHi infection…’ What effect specifically! The figure legend is confusing, lack detail and does not adequately describe the data. 4b – there is no effect on bacterial load. This is a single timepoint. Certainly a timecourse is necessary and might reveal reduced bacterial load/clearance with FliC treatment.

4F data does indicate that FliC reduced bacterial load is mediated by IL-22 (again exact numbers per group need to be shown). Fig 4e does not appear that reduced bacterial load is associated with increased numbers in the lung of a particular immune cell population. You should avoid reporting data using subject language such as ‘slightly increase numbers…’ The conclusion for this data relates to protective effect of flagellin during COPD exacerbation – this is not accurate.

Fig 5 for the most part FliC treatment reduced AMP expression with the exception of Defb2 gene at 24 h in CS-exposed mice.

Reviewer #2: Perez-Cruz and colleagues present a study in which they test the effect of systemic flagellin treatment on the ability of mice exposed to air or smoke to clear NTHi, and the effects on associated histopathology and immune response. The topic, use of innate immune modulators in a therapeutic setting, is of interest, and certainly relevant in COPD where bacterial infections resulting in exacerbation are a significant clinical issue. While there is merit to the study overall, I found the study a little disjointed and hard to follow, the methods inadequately described and some of the conclusions drawn by the authors to be misleading. Authors should attempt to address comments should be addressed prior to publication in PLOS ONE or elsewhere.

Major comments:

1. Stats section lists use of Mann-Whitney for pairwise comparisons. This is inappropriate given the comparisons between 4 or 6 experimental groups in most figures. Non-parametric test like Kruskal-Wallis should be applied followed by pairwise comparisons that are corrected for multiple comparisons by some method such as Dunn’s or Bonferoni.

2. Inaccuracy of abstract. The abstract claims two different modes of intervention with flagellin were trialled – “preventive and therapeutic”. This statement is misleading as the authors only applied flagellin at one time point, immediately prior to bacterial infection, so should perhaps best be described as “prophylactic”. Authors also make claims about defensinb2 peptide production that are not demonstrated in the results.

3. Number of mice per group per experiment – can the authors provide exact n for each group in each figure or individual figure panels – current descriptions are a bit vague and don’t give these details at sufficient level.

Further comments:

4. Grammar and spelling require some attention – for example in abstract “According our preventative or therapeutic protocol, flagellin was administered intraperitoneally” - perhaps this should read “Flagellin was administered intraperitoneally in preventive or therapeutic treatment protocols.” or something similar. Other examples “Acute exacerbations invariably scarred the chronic course of COPD 9.” There are quite a lot of grammatical and spelling errors throughout the manuscript, careful copy editing required.

5. Reference list needs to be carefully checked and updated – for example reference 14 is a paper published in 2016, but appears as ‘in press’ in the reference list.

6. What was the status of mice purchased, were they SPF?

7. Please detail briefly mention method of CS exposure in methods (whole body, nose only? Primary or secondary smoke?)

8. Methods section only appears to list one time at which flagellin was administered (just prior to bacterial challenge), while the abstract refers to both preventive and therapeutic administration protocols – please make exactly clear what you mean by preventive and therapeutic administration in the method section. Therapeutic administration in mouse models refers to administration of an intervention after the insult, or after development of pathology.

9. Details of the IL22 knockout mouse experiment are sparse. What strain were these mice on? Were appropriate WT controls employed? Statement in methods is IL22-/- mice were infected or not with NTHi – does this mean no PBS control was used? What about CS exposure? Were IL22 knockout mice also male and 6-8 weeks of age? More clarity needed in methods.

10. Flow cytometry method, incomplete sentence “gating strategies are.” . Can example gating strategies be shown?

11. How were lung cells dissociated? Clarify in methods.

12. Histology scoring – cumulative score of up to 30 doesn’t make sense looking at table 4 – max score possible is 28. Also how many high power fields were quantified per lung? Has this scoring system been published elsewhere? please reference

13. Results page 11: “mice chronically exposed to CS developed the major COPD features” – what were these features and can you provide evidence of this? For example in graph 1d for PBS treated mice, there is no evidence of CS-induced increase in total BAL cell count, or neutrophil count, which seems unusual for a 12 week CS exposure and is not consistent with the induction of COPD-like features. Was histopathology score modified by CS vs ambient air alone?

14. Results page 11: “This increase was consistent at 48h p.i. for the total

` cell number and the neutrophil count (Additional figure 1c and not shown, p<0.01).” – do the authors mean supplemental figure 1c? Also I don’t think in the era of supplemental figures, that the authors should be referring to data not shown – please supply in supplemental figures – this comment also applies to other instances of data not shown in results.

15. Results page 13 – Figure 3: “To further analyze the potential of lung immune cells to promote efficient antibacterial immune response, these cells were restimulated ex vivo with heat-killed (HK) bacteria and their cytokine profiles were assessed.” Are these experiements conducted on total lung cell suspensions, or isolated cell populations from lungs? Unclear if differences in cytokine production result from differential responses of cells in suspension or due to differential make up of lung cell suspensions tested. This data seems hard to make any meaningful interpretation from as it stands.

16. Figure 3: graph axes should be better labelled so the figure is easier for the reader to interpret (e.g. pg/ml CYTOKINE X in TISSUE X)

17. Figure 4A-D: schematic lists either WT or IL22-/- mice are infected, but data in panel b it is not clear if WT or knockout data are presented – both should be included in the manuscript. Again this is another incidence of ‘data not shown’. Panel C, it is not clear what mice the two graphs refer to from figure or figure legend – do they represent WT vs knockout mice? Different time points? Different tissues? Figure 4d is not labelled in the figure, and again should include both WT and knockout for comparison.

18. Figure 4A-D: not clear why authors shift to a non-CS exposure setting for IL22 knockout experiments. Surely it would be more informative and fitting with the aims of this research paper to investigate WT vs IL22 KO in the setting of CS exposure and FLIC protective effects as has been done in the antibody blocking experiment?

19. Figure 4F-G: The effect of anti IL22 on bacterial load is clear, yet the effects on inflammatory cells in the lungs and BAL are minimal – how do the authors reconcile this apparent discrepancy?

20. Fig 5: label all graphs with air vs CS for clarity.

21. Figure 5: Can authors confirm defensinb2 mRNA result with protein measurement? The authors claim in their abstract that “Flagellin treatment also amplified the

production of the β-defensin2 anti-bacterial peptides.”. Based off the data presented this statement is misleading and should be revised. Without protein data the authors should not over-interpret this result as the data is limited.

6. PLOS authors have the option to publish the peer review history of their article (what does this mean?). If published, this will include your full peer review and any attached files.

Reviewer #1: No

Reviewer #2: No

---

## [Author Response · Author response to Decision Letter 0]

27 Jan 2021

Additional Editor Comments: 

Both reviewers raised significant issues that must be considered and addressed carefully to allow further consideration of publication. Please note that methodological rigour is a prerequisite for publication in PloS One and therefore it is essential that comments related to these (group sizes, appropriate control groups, replication of findings) are addressed adequately.

Journal Requirements:

2. During our internal review, we noticed you have overlapping text with the dissertation of one of the authors published here: 

https://tel.archives-ouvertes.fr/tel-01916088/document

Please clarify what type of copyright, if any, the dissertation has and who the copyright owner is. PLOS ONE cannot (re)publish material without sufficient permission from the original copyright holder to publish under a CC BY license. If the copyright is not owned by the author and is not under a under a CC BY license, please complete one of the following:

Please provide proof that the owner of the content (a) has given you written permission to use it, and (b) has approved of the CC BY license being applied to their content. You may have the following form completed by the owner as proof: https://journals.plos.org/plosone/s/file?id=7c09/content-permission-form.pdf.

Alternatively, you may electronically request permissions through from the copyright holder and send us proof of approval, as long as the approval clearly shows that the owner has approved of the CC BY license being applied to their content. Please see https://journals.plos.org/plosone/s/licenses-and-copyright for more information.

This archive includes the dissertation of Dr Bachirou Kone (one of our authors) PhD thesis and summarized the results presented in this article. Indeed, this research belonged to his PhD program. Moreover, the owner of this archive is the university of Lille, one of our institution. All the authors approved the copyright transfert to Plos One editor.

 We have now included the results previously reported as data not shown in the text or as supplementary information. In some cases, we have removed these data if they are not essential to our demonstration.

.

4. Your ethics statement must appear in the Methods section of your manuscript. If your ethics statement is written in any section besides the Methods, please move it to the Methods section and delete it from any other section. Please also ensure that your ethics statement is included in your manuscript, as the ethics section of your online submission will not be published alongside your manuscript.

 This has been done in the revised version.

Reviewers' comments:

Comments to the Author

1. Is the manuscript technically sound, and do the data support the conclusions?

Reviewer #1: No

Reviewer #2: Partly

2. Has the statistical analysis been performed appropriately and rigorously? 

Reviewer #1: No

Reviewer #2: No

We have now extensively reviewed the methods and the results in order to clarify our results and their interpretation. Moreover, we have also modified the statistical analysis.

3. Have the authors made all data underlying the findings in their manuscript fully available?

Reviewer #1: No

Reviewer #2: No

All the data are now deposited as supplementary data.

4. Is the manuscript presented in an intelligible fashion and written in standard English?

Reviewer #1: No

Reviewer #2: No

5. Review Comments to the Author

Reviewer #1: 

The manuscript describes use of a mouse model of chronic (12 week) CS-exposure induced lung disease directly followed by NTHi challenge.

The title is confusing – ‘…..haemophilus influenzae induced exacerbations in CS-exposed mice’. Exacerbations of what…? It suggests that there is a pre-existing lung disease and NTHi infection is making this worse ie exacerbation. However as I read on there is no evidence of established (12 week CS-induced) lung disease. So I think the title needs re-thinking to better reflect the model. FLiC is given systemically (via intraperitoneal injection) – what is the rationale for this (to treat COPD exacerbation)? Have the authors tried targeted (airway) treatment?

Although our model mimicks most of the clinical situation observed during COPD, we cannot call it as COPD. It is the reason why we used the mention “CS-exposd mice”. In order to avoid confusion, we have removed the exacerbation and changed the title as “The Toll-Like Receptor 5 agonist flagellin prevents non-typeable Haemophilus influenzae-induced lung infection in cigarette smoke-exposed mice”

We have previously showed in our murine model of chronic exposure to CS that mice exhibit chronic inflammatory response associated with airway remodeling (peribronchial inflammation including metaplasia of bronchial epithelium), impaired bacterial clearance and parenchymal destruction in the lungs, further culminating in irreversible airflow limitation (ref 4). This model reproduces the most important characteristics of COPD as mentioned in this reference. 

The pathophysiology of COPD exacerbation episodes implicated a defect in IL-22 production and a deleterious effect of neutrophils on lung function. Moreover, local administration of FliC in the lung promotes a strong neutrophil recruitment in the airways with a moderate effect on IL-22 secretion as compared to the intraperitoneal route. For these reasons, we first tested the intraperitoneal route although we also demonstrated that local administration was also efficient on the bacterial clearance of Streptococcus pneumoniae (ref 25).

Results

The first statement – ‘mice chronically exposed to CS developed the major COPD features’. What is the evidence for this? Simply providing a reference to another paper as evidence that CS exposure induced COPD features (which are not defined…) is not adequate. Based on title, abstract, intro this study appears (although a bit vague eg title) about bacterial exacerbation of COPD. Figs 1b and 1c present BAL and lung bacterial load data. Note the labelling is incorrect in figure legend. The results state levels of NTHi in the blood were measured – there is no data for this. The right panel for fig 1c indicates increased levels of bacteria in lung tissue with FliC treatment?

We have previously validated these data by using the same protocol of 12 weeks exposure in the reference 4 (Pichavant et al. 2014). These mice developed a mild COPD phenotype including chronic inflammatory response associated with airway remodeling, impaired bacterial clearance and parenchymal destruction in the lungs, further culminating in irreversible airflow limitation. We always observed these phenotype in CS-exposed mice.

We have modified the figure 1 according to your comment. As confirmed by the new statistical analysis (one way anova analysis (Kruskal Wallis test) followed by Dunn’s multiple comparison test), treatment with FliC did not significantly increased the bacterial load in lung tissue.

Please show individual data points for each graph -are you showing n = 4 mice for one experiment or combining three repeats to show n = 12 per treatment? If n = 4 data is analysed this is not parametric and therefore mean +/-SEM not appropriate.

Fig 1D does include a PBS treated group to enable comparison of CS vs air treated mice. There is no evidence of airway (BAL) inflammation. Airway inflammation woul be considered a ‘major feature of COPD’. Why do the authors think CS exposure has not caused airway inflammation? Fig 1E – no PBS group here so cannot determine if CS exposure (alone) vs air (alone) has caused any COPD-like disease such as alveolar destruction as measured by mean linear intercept via quantitation of lung histology.

The data in figure 1 is not consistent with a model of bacterial COPD exacerbation since there are essentially no disease outcomes apparent in the CS (PBS) group vs Air (PBS) group. Ie no evidence that CS exposure has caused disease. The lung disease outcomes presented are driven by NTHi. The possibility that prior CS exposure has modified susceptibility to NTHi induced lung disease. However, no statistical comparison of CS (NTHi) vs Air (NTHi) has been conducted so there is no evidence that CS exposure has modified response to NTHi.

Fig 1E – there is no CS (PBS) and Air (PBS) group so cannot assess CS-induced lung disease.

The figures have been modified in order to include data points. This is not the topic of our article to demonstrate that CS induced features mimicking COPD and we have previously reported these data with the same experimental model with and without NTHi infection (ref 4 and 14). We also confirmed in the figure 1 that CS increased the bacterial load in CS-exposed mice and the inflammatory cell recruitment as previously reported (ref 14). We have added the data as requested. 

Fig 2 Unclear how many data points per treatment have been analysed. Please show individual data points – particularly if some groups have 3 mice so cannot determine if stats are appropriate… It is not clear to this reviewer what the key results here. Yes the different treatments are modifying cytokine expression – but the relevance of this to a specific pathway, disease mechanisms is not clear. Several sentences begin with ‘interestingly’ - interesting perhaps.. but is the relevance to bacterial COPD exacerbation, particularly given that this is not really modelling COPD exacerbation (no evidence of COPD-like disease)

We have now included the figures including each experimental points representing one mice. As mentioned in the introduction the Th1 and Th17 cytokines are both implicated in the COPD physiopathology and in the anti-bacterial response (including NTHi). This explains the fact that we have analyzed their production and of cytokines controlling their secretion such as IL-12p70, IL-1β, IL-6 and IL-23.

Fig 3. A lot of sup data or data not shown. Not clear why you are looking at immune cells in spleen when this a study of lung infection/disease. 3A cytokine production by spleen cells from CS-exposed mice +/- NTHi +/- FliC. Again don’t know number of data points for this data, and not statistically significant so does not add anything. 3B – why is this relevant and how does it inform on pathogenesis bacterial COPD exacerbation?

Since the mice are treated by intraperitoneal route, we can hypothesize that this treatment first affect the immune response within the spleen. After this, we suspect that the modulation of the splenic immune response might have some impact within the lung through the recirculation of immune mediators or cells. The aim was not to elucidate the pathogenesis of COPD exacerbation, we have previously done this in the references 13 and 14, but to determine how flagellin might prevent these episodes. We can hypothesize that flagellin acts by inducing the circulation of Th1 and/or Th17 cells or mediators. 

Fig 4 ‘to identify the role of IL-22 in the effect of flagellin during NTHi infection…’ What effect specifically! The figure legend is confusing, lack detail and does not adequately describe the data. 4b – there is no effect on bacterial load. This is a single timepoint. Certainly a timecourse is necessary and might reveal reduced bacterial load/clearance with FliC treatment.

4F data does indicate that FliC reduced bacterial load is mediated by IL-22 (again exact numbers per group need to be shown). Fig 4e does not appear that reduced bacterial load is associated with increased numbers in the lung of a particular immune cell population. You should avoid reporting data using subject language such as ‘slightly increase numbers…’ The conclusion for this data relates to protective effect of flagellin during COPD exacerbation – this is not accurate.

Our data suggest that flagellin mostly prevent the NTHi-induced exacerbation of COPD by promoting the bacterial clearance and by decreasing the lung inflammatory reaction. So our aim in the figure 4 is to determine the role of IL-22 on both parameters. We have chosen this time points since we have previously showed that flagellin had a significant effect at day 2 and to efficiently addressed the impact on lung alterations and inflammation. We have modified the figure according to your requests and the related comments within the result sections has been clarified. 

Fig 5 for the most part FliC treatment reduced AMP expression with the exception of Defb2 gene at 24 h in CS-exposed mice.th

We agree with the comment of the reviewer. Our explanation for this effect is that treatment with flagellin accelerates the bacterial clearance within the lung and by this way, decreases the bacteria-induced AMP production. In this article, we also show that this treatment reduces the recruitment of effector cells such as macrophages and neutrophils. This might be the reflect of the same mechanism as mentioned in the discussion. 

Reviewer #2: 

Perez-Cruz and colleagues present a study in which they test the effect of systemic flagellin treatment on the ability of mice exposed to air or smoke to clear NTHi, and the effects on associated histopathology and immune response. The topic, use of innate immune modulators in a therapeutic setting, is of interest, and certainly relevant in COPD where bacterial infections resulting in exacerbation are a significant clinical issue. While there is merit to the study overall, I found the study a little disjointed and hard to follow, the methods inadequately described and some of the conclusions drawn by the authors to be misleading. Authors should attempt to address comments should be addressed prior to publication in PLOS ONE or elsewhere.

We would like to thank the reviewer for his/her positive comments.

Major comments:

1. Stats section lists use of Mann-Whitney for pairwise comparisons. This is inappropriate given the comparisons between 4 or 6 experimental groups in most figures. Non-parametric test like Kruskal-Wallis should be applied followed by pairwise comparisons that are corrected for multiple comparisons by some method such as Dunn’s or Bonferoni.

We have now re-analyzed our data by using one way ANOVA analysis (Kruskal Wallis test) followed by Dunn’s multiple comparison test. Our results section has been changed according to these analysis.

2. Inaccuracy of abstract. The abstract claims two different modes of intervention with flagellin were trialled – “preventive and therapeutic”. This statement is misleading as the authors only applied flagellin at one time point, immediately prior to bacterial infection, so should perhaps best be described as “prophylactic”. Authors also make claims about defensinb2 peptide production that are not demonstrated in the results.

We have now modified the abstract according to your comments and we have measured the concentrations of defensin-β2. These data showed that the concentrations of Defensin-β2 were higher in BALF from Air- and CS-exposed mice infected with NTHi as compared to controls. However, treatment with FliC did not upregulated these levels both in the BAL and the lung protein extract. We have modified the result section and the figure 4 in order to include these data as well as our interpretation of these results. Accordingly, we have removed the mention that upregulation of defb2 mRNA can mediate the effect of flagellin. However, we cannot excluded that flagellin upregulates the expression of others AMP. We have also changed preventive for prophylactic.

3. Number of mice per group per experiment – can the authors provide exact n for each group in each figure or individual figure panels – current descriptions are a bit vague and don’t give these details at sufficient level.

As required by the reviewer 1, we have now added the individual points in each figure and the number of mice and/or of experiments was clarified. 

Further comments:

4. Grammar and spelling require some attention – for example in abstract “According our preventative or therapeutic protocol, flagellin was administered intraperitoneally” - perhaps this should read “Flagellin was administered intraperitoneally in preventive or therapeutic treatment protocols.” or something similar. Other examples “Acute exacerbations invariably scarred the chronic course of COPD 9.” There are quite a lot of grammatical and spelling errors throughout the manuscript, careful copy editing required.

Our manuscript has been carefully edited by an expert in English. 

5. Reference list needs to be carefully checked and updated – for example reference 14 is a paper published in 2016, but appears as ‘in press’ in the reference list.

We have modified this reference and controlled the other ones.

6. What was the status of mice purchased, were they SPF?

All the mice used for our experiments are SPF. The mice were acclimated in our house facility during at least one week before to start our experimental protocol. 

7. Please detail briefly mention method of CS exposure in methods (whole body, nose only? Primary or secondary smoke?)

The mice were exposed to primary cigarette smoke in a whole body chamber. We are using the Inexpose system from EMKA (Paris- France). This has been included in the methods section.

8. Methods section only appears to list one time at which flagellin was administered (just prior to bacterial challenge), while the abstract refers to both preventive and therapeutic administration protocols – please make exactly clear what you mean by preventive and therapeutic administration in the method section. Therapeutic administration in mouse models refers to administration of an intervention after the insult, or after development of pathology.

We have now indicated in the material and methods the specificities of both protocols (last paragraph of Mice infection and flagellin administration). 

9. Details of the IL22 knockout mouse experiment are sparse. What strain were these mice on? Were appropriate WT controls employed? Statement in methods is IL22-/- mice were infected or not with NTHi – does this mean no PBS control was used? What about CS exposure? Were IL22 knockout mice also male and 6-8 weeks of age? More clarity needed in methods.

WT and IL-22-/- mice have a C57BL/6J genetic background. Both are acclimated to our animal facility for at least one week before to start the protocols. In IL-22-/- mice, we used male mice and not-infected mice received PBS. These KO mice were not exposed to CS since it has been reported that a defect in IL-22 impaired the development of lung disease in CS-exposed mice (Starkey MR et al, ERJ. 2019). We have precised in the revised version of our manuscript the origin of our KO mice and the conditions of the mice housing.

10. Flow cytometry method, incomplete sentence “gating strategies are.” . Can example gating strategies be shown?

Our gating strategy has been previously reported in the reference 14 from Sharan R. et al. We have added the reference in the section “flow cytometry”. 

11. How were lung cells dissociated? Clarify in methods.

Lungs were perfused with PBS and right lobe of lung was treated with collagenase (Sigma-Aldrich). The leucocyte-enriched fraction was collected using a Percoll gradient (GE Healthcare) before flow cytometry staining and culture. This has been added in the « Sample collection and processing » section.

12. Histology scoring – cumulative score of up to 30 doesn’t make sense looking at table 4 – max score possible is 28. Also how many high power fields were quantified per lung? Has this scoring system been published elsewhere? please reference

Indeed, we have made a mistake in this table. We have defined two parameters in order to measure the vasculitis, either endothelium necrosis or the presence of inflammatory cell recruitment around the blood vessel and one is missing in the first version of our article. We have modified the table in the revised version and the total of the score is in fact of 30. We have analyzed at least 5 high power fields and this scoring has been recently published (ref 25). 

13. Results page 11: “mice chronically exposed to CS developed the major COPD features” – what were these features and can you provide evidence of this? For example in graph 1d for PBS treated mice, there is no evidence of CS-induced increase in total BAL cell count, or neutrophil count, which seems unusual for a 12 week CS exposure and is not consistent with the induction of COPD-like features. Was histopathology score modified by CS vs ambient air alone?

In order to complete the histologic analysis in our article, we have now measured the mean linear intercept (MLI) in lung sections of our mice (Sup table X). Mice exposed to CS exhibit a significantly higher histologic score and MLI as compared to mice exposed to air as previously reported (ref pichavant). Regarding the number of BAL total and neutrophil counts in control Air- and CS-exposed mice, the lack of difference is probably related to the fact that mice were no more exposed to CS during the infection protocol since they were transferred in a A2 animal facility for this. The sacrifices were performed 4-5 days (day 1 or 2 after infection) after the last exposure to CS.

14. Results page 11: “This increase was consistent at 48h p.i. for the total

` cell number and the neutrophil count (Additional figure 1c and not shown, p<0.01).” – do the authors mean supplemental figure 1c? Also I don’t think in the era of supplemental figures, that the authors should be referring to data not shown – please supply in supplemental figures – this comment also applies to other instances of data not shown in results.

In the revised version of our manuscript, we have removed the data not shown and we have added the most important ones in the text and the supplementary figures. The cell number at 48h p.i. are indeed reported in the figure 1c. 

15. Results page 13 – Figure 3: “To further analyze the potential of lung immune cells to promote efficient antibacterial immune response, these cells were restimulated ex vivo with heat-killed (HK) bacteria and their cytokine profiles were assessed.” Are these experiements conducted on total lung cell suspensions, or isolated cell populations from lungs? Unclear if differences in cytokine production result from differential responses of cells in suspension or due to differential make up of lung cell suspensions tested. This data seems hard to make any meaningful interpretation from as it stands.

In vitro cell activation was performed on total lung cell suspension. We cannot correlated the variation in cytokine production with significant modulation in percentages of leucocytes, T cell populations and not conventional T cells. Nevertheless, we cannot exclude this link and we have added a comment in the discussion concerning this potential link. 

16. Figure 3: graph axes should be better labelled so the figure is easier for the reader to interpret (e.g. pg/ml CYTOKINE X in TISSUE X)

We have now modified the labelling of the figures in order to facilitate their reading.

17. Figure 4A-D: schematic lists either WT or IL22-/- mice are infected, but data in panel b it is not clear if WT or knockout data are presented – both should be included in the manuscript. Again this is another incidence of ‘data not shown’. Panel C, it is not clear what mice the two graphs refer to from figure or figure legend – do they represent WT vs knockout mice? Different time points? Different tissues? Figure 4d is not labelled in the figure, and again should include both WT and knockout for comparison.

The legend of this figure and the comment in the result section has been modified in order to improve the reading of these graphs. As mentioned above, the data not shown has been removed and the data were reported in the text. 

18. Figure 4A-D: not clear why authors shift to a non-CS exposure setting for IL22 knockout experiments. Surely it would be more informative and fitting with the aims of this research paper to investigate WT vs IL22 KO in the setting of CS exposure and FLIC protective effects as has been done in the antibody blocking experiment?

The defect in IL-22 can also modify the response to CS as previously reported by Starkey et al (ref 17) . In this context, it will be difficult to compare the effect of infection by NTHi in CS-exposed WT and IL-22-/- mice if they exhibit a different response to CS and more specifically, different immune responses.

19. Figure 4F-G: The effect of anti IL22 on bacterial load is clear, yet the effects on inflammatory cells in the lungs and BAL are minimal – how do the authors reconcile this apparent discrepancy?

We suspect that IL-22 had no direct effect on the recruitment of inflammatory cells in contrast to its impact on bacterial clearance. In this model, the bacterial clearance was not related to the number of leucocytes. The priming of these cells is also essential in order to improve their ability to clear bacteria. 

20. Fig 5: label all graphs with air vs CS for clarity.

The figure has been modified according to your comment. 

21. Figure 5: Can authors confirm defensinb2 mRNA result with protein measurement? The authors claim in their abstract that “Flagellin treatment also amplified the

production of the β-defensin2 anti-bacterial peptides.”. Based off the data presented this statement is misleading and should be revised. Without protein data the authors should not over-interpret this result as the data is limited.

The measurement of defensing-b2 by ELISA showed that the concentrations were higher in BALF from Air- and CS-exposed mice infected with NTHi as compared to controls. However, treatment with FliC did not upregulated these levels both in the BAL and the lung protein extract. We have modified the result section and the figure 4 in order to include these data as well as our interpretation of these results. Accordingly, we have removed the mention that upregulation of defb2 mRNA can mediate the effect of flagellin.

6. PLOS authors have the option to publish the peer review history of their article (what does this mean?). If published, this will include your full peer review and any attached files.

Do you want your identity to be public for this peer review? For information about this choice, including consent withdrawal, please see our Privacy Policy.

Reviewer #1: No

Reviewer #2: No

---

## [Decision Letter · Decision Letter 1]

26 Feb 2021

The Toll-Like Receptor 5 agonist flagellin prevents Non-typeable Haemophilus influenzae-induced infection in cigarette smoke-exposed mice.

PONE-D-20-19459R1

Dear Dr. Gosset,

We’re pleased to inform you that your manuscript has been judged scientifically suitable for publication and will be formally accepted for publication once it meets all outstanding technical requirements.

Kind regards,

Aran Singanayagam

Academic Editor

PLOS ONE

Additional Editor Comments (optional):

Reviewers' comments:

Reviewer's Responses to Questions

**Comments to the Author**

1. If the authors have adequately addressed your comments raised in a previous round of review and you feel that this manuscript is now acceptable for publication, you may indicate that here to bypass the “Comments to the Author” section, enter your conflict of interest statement in the “Confidential to Editor” section, and submit your "Accept" recommendation.

Reviewer #2: All comments have been addressed

2. Is the manuscript technically sound, and do the data support the conclusions?

Reviewer #2: Yes

3. Has the statistical analysis been performed appropriately and rigorously? 

Reviewer #2: Yes

4. Have the authors made all data underlying the findings in their manuscript fully available?

Reviewer #2: Yes

5. Is the manuscript presented in an intelligible fashion and written in standard English?

Reviewer #2: No

6. Review Comments to the Author

Reviewer #2: The authors have addressed my comments, although there remain numerous spelling and grammatical errors in the text.

7. PLOS authors have the option to publish the peer review history of their article (what does this mean?). If published, this will include your full peer review and any attached files.

Reviewer #2: No

---

## [Editor Report · Acceptance letter]

18 Mar 2021

PONE-D-20-19459R1 

The Toll-Like Receptor 5 agonist flagellin prevents *Non-typeable Haemophilus influenzae*-induced infection in cigarette smoke-exposed mice. 

Dear Dr. Gosset:

I'm pleased to inform you that your manuscript has been deemed suitable for publication in PLOS ONE. Congratulations! Your manuscript is now with our production department. 

Kind regards, 

on behalf of

Dr. Aran Singanayagam 

Academic Editor

PLOS ONE